# Estimating Gradients of Physical Fields in Space

Yufei Zhou and Chao Shen

Harbin Institute of Technology (Shenzhen), Shenzhen, Guangdong, China

**Correspondence:** Chao Shen (shenchao@hit.edu.cn)

**Abstract.** This study focuses on the development of a multipoint technique for future constellation missions, aiming to measure gradients at various order, in particular the linear and quadratic gradients, of a general field. It is well-established that in order to estimate linear gradients, the spacecraft must not lie on a plane. Through analytical exploration within the framework of least-squares, it is demonstrated that at least ten spacecraft that do not lie on any quadric surface are required to estimate both linear and quadratic gradients. The spatial arrangement of the spacecraft can be characterized by a set of quality factors. In cases where there is poor temporal synchronization among the spacecraft, leading to non-simultaneous measurements, temporal gradients must be included. If the spacecraft have multiple velocities, by incorporating temporal gradients it is possible to reduce the number of required spacecraft. Furthermore, it is proved that the accuracy of the linear gradient is of second order and that of the quadratic gradient is of first order. Additionally, a method for estimating errors in the calculation is also illustrated.

## 1 Introduction

Multipoint measurements have significantly advanced our understanding of the structures and dynamics of the space plasmas. The basic approach involves direct interpretation of the collected data. However, to maximize the potential of these measurements, several techniques have been developed to estimate additional quantities that would otherwise remain inaccessible. One common initial step is to estimate the linear gradients of physical fields, with particular focus on the magnetic field (Chanteur, 1998; De Keyser, 2008; De Keyser et al., 2007; Dunlop et al., 1988; Harvey, 1998; Hamrin et al., 2008; J. Vogt et al., 2008; Vogt et al., 2009; Shen and Dunlop, 2023). These gradients serve various purposes, such as calculating the electric current density (Dunlop et al., 2015, 2016, 2018), determining the curvature and rotation rate of magnetic field lines (Shen et al., 2003, 2007), locating magnetic nulls crucial for magnetic reconnection (Fu et al., 2015), and determining the dimensionality and velocity of magnetic structures (Shi et al., 2005, 2006; Fadanelli et al., 2019). A recent technique utilizes the gradients of normal fields on curved boundary layers to estimate the principal curvatures and directions of the boundary layers (Shen et al., 2020; Shao et al., 2023; Zhou et al., 2023).

The recent MMS (Magnetospheric Multiscale) mission has improved particle data measurements with exceptional resolution. With this capability, the electric current can be directly calculated by summing the product of the bulk flux and charge of particles (Burch et al., 2015; Pollock et al., 2016). By leveraging Maxwell's equations and incorporating additional information, such as the electric current measurements from each spacecraft, it becomes possible to estimate not only the linear gradients but also the quadratic gradients of magnetic fields using four-point measurement (Liu et al., 2019; Torbert et al., 2020; Denton

et al., 2020; Shen et al., 2021a), though the general estimation of these gradients of an arbitrary field typically requires ten spacecraft measurement (Chanteur, 1998; Shen et al., 2021c). With both linear and quadratic gradients known, the complete geometry of the magnetic field lines, including their curvature and torsion, can be obtained (Shen et al., 2021a; Torbert et al., 2020). This is of particular use in the reconstruction of key regions such as in reconnection. Unlike other reconstruction methods (see, e.g. Sonnerup and Teh, 2008; Hasegawa et al., 2021), the approach utilizing gradients avoids assumptions specific to the reconnection process, thus making it adaptable to a wide range of conditions, though it is still limited by the separation of the spacecraft such that the result may not be as accurate when studying those phenomena of much smaller or larger scales.

At present, there is a growing tendency of enhanced resolution in particle and electric field measurement and increased number of spacecraft involved in a multispacecraft mission (Ogilvie et al., 1977; Escoubet et al., 2001; Liu et al., 2005; Friis-Christensen et al., 2006; Angelopoulos, 2008; Burch et al., 2015; Spence et al., 2022; Maruca et al., 2021). An algorithm for the linear and quadratic gradients (ALQG) has been developed that relies on ten or more measurement points to tackle the general problem of estimating quadratic gradients of physical fields that are not limited to magnetic fields alone (Shen et al., 2021c). In ALQG, the quadratic gradients can be obtained by solving a matrix equation. The characteristic matrix, $\Re^{MN}$, that is determined by the positions of the spacecraft within the constellations, has been put forward. If the determinant of the characteristic matrix $\Re^{MN}$ is non-zero, the full quadratic gradients can be obtained. One application is the measurement of electric charge density using the Poisson equation (Shen et al., 2021c, b). In this approach, the charge density is calculated by summing the diagonal elements of the estimated quadratic gradient matrix of a potential field (Shen et al., 2021b).

However, despite progress in addressing some of the associated challenges, several issues remain unresolved. The first problem revolves around the relationship between the feasibility of estimation and the distribution of measurement points. It is well-established that in four-point measurements, linear gradients can be obtained as long as the points do not lie on a plane (Vogt et al., 2009; Shen et al., 2012; Shen and Dunlop, 2023). However, the impact of the point distribution on the estimation of quadratic gradients has not been fully understood. This poses a challenge in determining the optimal distribution that ensures accurate estimation. When four spacecraft are on a plane, it is still possible to obtain the linear gradients in the plane (Vogt et al., 2009; Shen et al., 2012; Shen and Dunlop, 2023). When dealing with quadratic gradients, if a distribution of measurement points is found unsuitable for achieving a complete estimation, there is no method available to extract the utmost information regarding the gradients.

The second problem concerns the requirement of simultaneity in measurements, which applies to both the new technique for quadratic gradients and previous techniques for linear gradients (Harvey, 1998; Chanteur, 1998; Hamrin et al., 2008). As the number of spacecraft increases, the issue of temporal synchronization among them becomes more pronounced. One possible approach to mitigate this problem is to incorporate temporal gradients into the analysis (De Keyser et al., 2007; De Keyser, 2008).

The third problem pertains to the accuracy of the estimation process and the associated errors. Although the technique has demonstrated high accuracy when tested on synthetic data, with suggestions that errors in linear gradients are of second order and errors in quadratic gradients are of first order (Shen et al., 2021c), these results have not been deduced analytically.

In practical applications, measurements include noise which may also affect estimates of gradients. It is therefore crucial to develop a reliable method for estimating and quantifying errors of various origins.

In these regards, this study presents a further development to ALQG. In addition to calculating quadratic gradients, the results can also be applied to reconstruct physical fields and structures in space using polynomials.

## 2  The Problem

We start with the problem of approximation. To approximate a vector field, an approach is to aggregate the approximations of its individual component fields, treating each component as an independent scalar field. This method is useful when there is no additional information available regarding the relationship among the component fields, such as the constraint $\nabla \cdot \boldsymbol{B} = 0$. For simplicity we here consider the problem of approximating a scalar field in space, and the result can be applied equally well to vector fields.

A field can be seen as a combination of multiple constituent fields originating from different sources. These fields often have distinct temporal and spatial scales. For instance, in the inner magnetosphere during a substorm, the total magnetic field comprises the dipole (and higher-order moments) geomagnetic field, disturbances caused by currents (Yang et al., 2012), and other localized and temporary variations. On the bow shock front, various waves superimpose. In most cases, our focus is on specific constituents, such as the disturbance field during a substorm or the shock ramp on a shock front. Therefore, we can express the total field $j(\boldsymbol{x})$ as the sum of a background field of interest $f(\boldsymbol{x})$ and wave fields $w(\boldsymbol{x})$ with smaller scales compared to $f(\boldsymbol{x})$:

$$j(\boldsymbol{x}) = f(\boldsymbol{x}) + w(\boldsymbol{x}) \tag{1}$$

Here, $\boldsymbol{x}$ is a $r$-component vector representing a point. The general case is when $r = 4$ and $\boldsymbol{x} = [vt, x, y, z] = [x_0, x_1, x_2, x_3]$, which represents a point in time-space. $v$ is a constant to scale the temporal coordinate with the spatial ones. It can be chosen as the characteristic speed in the system, such as the mean Alfvén speed or flow speed in the region of concern. The choice of $v$ does not impact the general method described below. The scaling of spatial coordinates are discussed in Appendix B. It is also possible to consider the field in a cut of time-space, that is, at a specific time. Then $r = 3$ and $\boldsymbol{x} = [x, y, z] = [x_1, x_2, x_3]$ represents a point in space. Since our objective is to approximate $f(\boldsymbol{x})$, we can represent it using multi-index notation (see Appendix A) as a sum of multivariate polynomials:

$$f(\boldsymbol{x}) = \sum_{|\alpha|=0}^{\infty} g_\alpha \boldsymbol{x}^\alpha. \tag{2}$$

In this equation, $g_\alpha$ is the coefficient of the polynomial $\boldsymbol{x}^\alpha$, and we employ the properties of multi-index notation (Properties 3 and 4 in Appendix A). By comparing Equation (2) with the Taylor expansion of $f(\boldsymbol{x})$ around $\bar{0} = [0,0,0,0]$, we can see that the coefficients $g_\alpha$ are related to the gradients $f_{,\alpha}(\bar{0})$ as follows:

$$g_\alpha = \frac{f_{,\alpha}(\bar{0})}{\alpha!} \tag{3}$$

where we employ Property 8 of multi-index notation. Suppose we aim to approximate $f(\boldsymbol{x})$ using polynomials up to degree $d$. We define:

$$p_d(\boldsymbol{x}) \equiv \sum_{|\alpha|=0}^{d} g_\alpha \boldsymbol{x}^\alpha, \tag{4}$$

$$p_d^+(\boldsymbol{x}) \equiv \sum_{|\alpha|=d+1}^{\infty} g_\alpha \boldsymbol{x}^\alpha, \tag{5}$$

By doing so, we separate the summation in Equation (2) into a polynomial of degree at most $d$, denoted as $p_d(\boldsymbol{x})$, and a polynomial in which all terms have degrees higher than $d$, denoted as $p_d^+(\boldsymbol{x})$. There are $\binom{d+r}{r} = (d+r)!/r!d!$ terms in Equation (4), resulting in an equal number of coefficients to be determined from measured data. Now we can rewrite Eq. (1) as

$$j(\boldsymbol{x}) = p_d(\boldsymbol{x}) + p_d^+(\boldsymbol{x}) + w(\boldsymbol{x}) \tag{6}$$

When field measurements are conducted using probes, we need to consider the positioning error in time-space, denoted
as $\delta\boldsymbol{x} = [v\delta t, \delta x, \delta y, \delta z]$. Suppose we think the total field is measured at $\boldsymbol{x}_m$, but due to the positioning error, it is actually measured at $\boldsymbol{x}_m + \delta\boldsymbol{x}$. Taking into account the measurement error in the field, denoted as $\delta j$, we can express the sampled data $j_m$ as follows:

$$j_m = p_d(\boldsymbol{x}_m + \delta\boldsymbol{x}) + p_d^+(\boldsymbol{x}_m + \delta\boldsymbol{x}) + w(\boldsymbol{x}_m + \delta\boldsymbol{x}) + \delta j \tag{7}$$

Note that the scaling factor $v$ for the temporal coordinate $x_{m0} = vt_m$ is the same for all measurement points.
Consider $M$ measurements taken at different points in time-space, yielding data pairs $j_m$ and $\boldsymbol{x}_m$ for $1 \le m \le M$. The objective is to determine a set of numerical values for $g_\alpha$, where $|\alpha| \le d$, that yield the best approximation of $f(\boldsymbol{x})$ by $p_d(\boldsymbol{x})$ based on this data. It is evident from Equation (7) that the discarded polynomial $p_d^+(\boldsymbol{x})$, the wave field $w(\boldsymbol{x})$, the measurement error $\delta j$, and the positioning error $\delta\boldsymbol{x}$ all contribute to the final error when solving this problem. The concept of measurement by probes in space is illustrated in Fig. 1.

## 3   The Solution

We define the error between the measured field and the approximating polynomial as $s_m$, given by:

$$s_m = j_m - p_d(\boldsymbol{x}_m). \tag{8}$$

To quantify the total error, we employ the weighted least-square method, which constructs the total error as a weighted sum of all individual errors:

$$S = \sum_{m,n}^{M} s_m W_{mn} s_n, \tag{9}$$

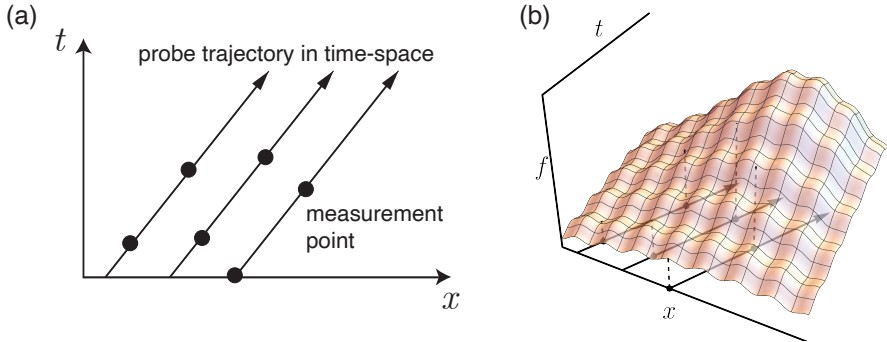

**Figure 1.** Multi-spacecraft measurement in time-space. (a) The trajectories of multiple probes/spacecraft that fly in formation in time-space are represented by arrows. (b) Measurements of a scalar field $f$ are made along the trajectories.

Here, the weight matrix $W_{mn}$ determines the contribution of each measurement to the approximation. The choice of the weight matrix depends on the specific problem (De Keyser et al., 2007), but in a simple case where all measurements are equally important, it can be expressed as:

$$W_{mn} = \delta_{mn}/M. \tag{10}$$

Generally, it is symmetric and invertible. Minimization of the total error with respect to $g_\beta$ and assuming that this is done when $g_\beta = \tilde{g}_\beta$, result in a set of $\binom{d+r}{r}$ equations for $\tilde{g}_\beta$:

$$\left.\frac{\partial S}{\partial g_\beta}\right|_{\tilde{g}_\beta} = 0. \tag{11}$$

We define the matrix $\mathbf{R}$ with elements:

$$R_{\beta\alpha} \equiv \sum_{m,n}^{M} \boldsymbol{x}_m^\beta W_{mn} \boldsymbol{x}_n^\alpha, \tag{12}$$

Additionally, we define:

$$J_\beta \equiv \sum_{m,n}^{M} \boldsymbol{x}_m^\beta W_{mn} j_n. \tag{13}$$

With these notations, taking into account Equations (9), (8), (7), and (4), Equation (11) can be explicitly expressed as a system of equations:

$$J_\beta = \sum_{|\alpha| \leq d} R_{\beta\alpha} \tilde{g}_\alpha, \tag{14}$$

This linear system of equations consists of $\binom{d+r}{r}$ equations and unknowns. The tilde notation on $g_\alpha$ signifies that it represents an estimated quantity rather than the true value.

The solution to Equation (14), i.e., the estimation $\tilde{g}_\alpha$, can be obtained directly using common computer programs designed to solve linear systems. By applying the relation in Equation (3), the gradients up to the $d$th degree of the field at the origin $\bar{0}$ can be determined. The approximation of the field $f(\boldsymbol{x})$ is then given by:

$$\tilde{p}_d(\boldsymbol{x}) = \sum_{|\alpha| \leq d} \tilde{g}_\alpha \boldsymbol{x}^\alpha. \tag{15}$$

It is important to note that, at this stage, the coordinate system, specifically its origin, has not been chosen. In Section 5, we will demonstrate that the center of the measurement points, if chosen as the origin, yields the best reduction of the approximation error resulting from the truncation of the Taylor series.

## 4 Existence and Uniqueness of Solution and Implication for Multispacecraft Mission Design

### 4.1 The Requirement for a Unique Solution

From Eq. (14) there exists a unique set of solution for $\tilde{g}_\alpha$ if and only if $R$ has full rank. This requirement has several implications regarding the number, distribution, and velocity of probes in space. To see these we need to decompose $\mathbf{R}$.

Based on the decomposition of the symmetric and invertible weight matrix as $W_{mn} = \sum_{l,s,k}^{M} (P^T)_{ml} O_{ls} O_{sk} P_{kn}$, where $\mathbf{O}$ is a diagonal matrix whose elements, when squared, are the eigenvalues of the weight matrix and $\mathbf{P}$ is composed of eigenvectors, we can express the matrix $\mathbf{R}$ as:

$$R_{\beta\alpha} = \sum_{m,n,l,s,k}^{M} \boldsymbol{x}_m^\beta (P^T)_{ml} O_{ls} O_{sk} P_{kn} \boldsymbol{x}_n^\alpha, \tag{16}$$

Considering the relation $\text{rank}(\mathbf{A}^{\mathrm{T}}\mathbf{A}) = \text{rank}(\mathbf{A})$ and the invertibility of $\mathbf{OP}$, we have:

$$\text{rank}(\mathbf{R}) = \text{rank}(\mathbf{OPX}) = \text{rank}(\mathbf{X}) \tag{17}$$

where the matrix $\mathbf{X}$ is defined by

$$X_{n\alpha} \equiv \boldsymbol{x}_n^\alpha. \tag{18}$$

Therefore, the uniqueness of the solution in Equation (14) is equivalent to the rank of $\mathbf{X}$ being $\binom{d+r}{r}$.

The matrix $\mathbf{X}$ has rows corresponding to different measurement points and columns corresponding to coefficients $g_\alpha$. To achieve a rank of $\binom{d+r}{r}$ for $\mathbf{X}$, two conditions need to be met. First, the number of measurement points $M$ should be at least $\binom{d+r}{r}$. Second, the points should not all lie on an algebraic surface of degree at most $d$, ensuring that there is no set of coefficients $a_\alpha$ such that

$$\sum_{|\alpha| \leq d} a_\alpha \boldsymbol{x}_m^\alpha = 0. \tag{19}$$

A similar result has also been obtained for multivariate interpolations (Olver, 2006).

Although we present the first condition separately from the second to stress its utility in application, it is contained in the second since a lack of measurement points necessarily makes the existing points lie on a surface prescribed by the second. For example, three points ($d = 1, r = 3$) must be on a plane and nine points ($d = 2, r = 3$) must be on a second-order surface.

To illustrate the second condition we take $d = 2, r = 3$ as an example. The matrix $X$ in this case is given by

$$
\mathbf{X} = \begin{bmatrix}
1 & x_1 & y_1 & z_1 & x_1^2 & x_1 y_1 & x_1 z_1 & y_1^2 & y_1 z_1 & z_1^2 \\
1 & x_2 & y_2 & z_2 & x_2^2 & x_2 y_2 & x_2 z_2 & y_2^2 & y_2 z_2 & z_2^2 \\
1 & x_3 & y_3 & z_3 & x_3^2 & x_3 y_3 & x_3 z_3 & y_3^2 & y_3 z_3 & z_3^2 \\
\vdots & \vdots & \vdots & \vdots & \vdots & \vdots & \vdots & \vdots & \vdots & \vdots \\
1 & x_M & y_M & z_M & x_M^2 & x_M y_M & x_M z_M & y_M^2 & y_M z_M & z_M^2
\end{bmatrix}.
\tag{20}
$$

If all the points lie on a second-order algebraic surface, we can express the surface formally with appropriately chosen coefficients $a_\alpha$ as

$$
a_{(0,0,0)} + a_{(1,0,0)} x + a_{(0,1,0)} y + a_{(0,0,1)} z + a_{(2,0,0)} x^2 + a_{(1,1,0)} xy + a_{(1,0,1)} xz + a_{(0,2,0)} y^2 + a_{(0,1,1)} yz + a_{(0,0,2)} z^2 = 0
\tag{21}
$$

and all points satisfy this equation. This indicates that we can make a linear combination of the columns in Eq. (20) with the coefficients in Eq. (21) and obtain a column of zeros. Thus, the rank of $\mathbf{X}$ is lower than, $\binom{2+3}{3}$, the number of columns it possesses. On the other hand, if the points does not lie on a second-order algebraic surface, then there does not exist a set of coefficients to linearly combine the columns to reach a column of zeros. In this case the rank is $\binom{2+3}{3}$.

These two conditions have important implications for the orbit design of future multispacecraft missions and for adaptation of this general framework to specific problems in practice such as measuring electric charges (Shen et al., 2021b) and reconstructing magnetic structures (Liu et al., 2019; Torbert et al., 2020; Shen et al., 2021a). Here we discuss them in detail and illustrate with Fig. 2.

We first consider simultaneous measurements and $r = 3$. If $d = 1$, that is to estimate the spatial linear gradients, we recover the well-known restriction that at least four measurement points are needed, and these points should not lie on a first-order algebraic surface, or in other words a plane (Fig. 2 c), such that with appropriately chosen coefficients $a_\alpha$ they satisfy

$$
a_{(0,0,0)} + a_{(1,0,0)} x + a_{(0,1,0)} y + a_{(0,0,1)} z = 0,
\tag{22}
$$

If $d = 2$, in which case both the linear and quadratic gradients are to be estimated, we need at least ten measurement points. They should not reside on a second-order algebraic surface which can be defined by Eq. (21) (Fig. 2 f). Typical examples of second-order surface include ellipsoid, elliptic cone, elliptic cylinder, elliptic paraboloid. Among them the sphere is common as for the distribution of probes to date. The geomagnetic stations are on the surface of solid Earth. The Iridium satellite constellation are in the ionosphere.

Next we consider $r = 4$ and that time series data are incorporated to estimate the gradients of fields in time-space. If $d = 1$, at least five points are needed and they should not lie on a hypersurface in time-space. These five points can be obtained from

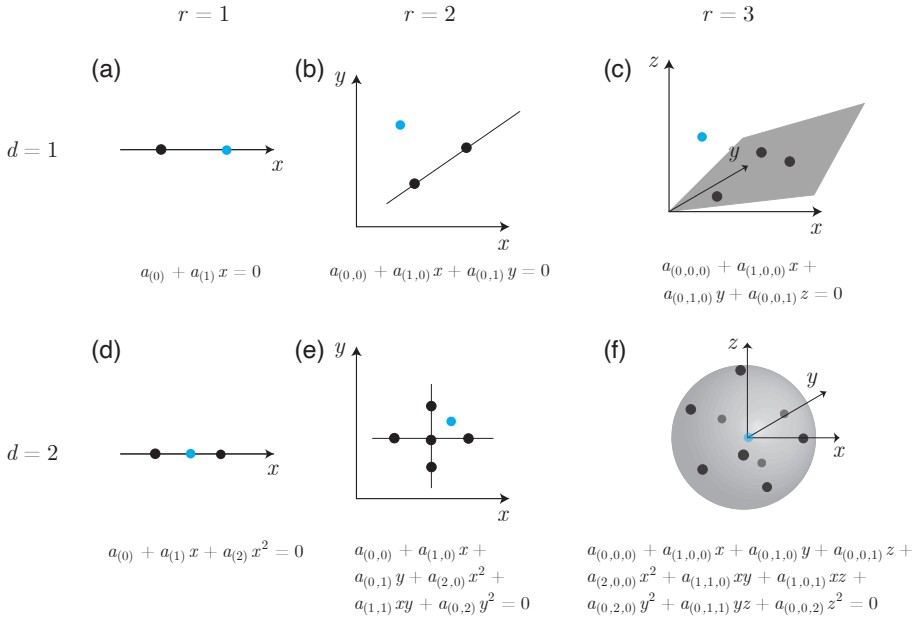

**Figure 2.** Distributional requirement of measurement points in estimating gradients up to first order ($d = 1$) in (a) one-dimensional ($r = 1$), (b) two-dimensional ($r = 2$), and (c) three-dimensional ($r = 3$) space, and gradients up tp second order in (d) one-dimensional, (e) two-dimensional, and (f) three-dimensional space. In each panel, the equation defines the hypersurface from which the distribution of measurement points should deviates; The black dots denote the measurement points that lie on some hypersurface defined by the equation with a particular set of coefficients. Such hypersurfaces are represented by the lines in panels (b) and (e), the plane in panel (c), the sphere in panel (f), and the dots in panels (a) and (d). The blue dots represent the additional measurement points not lying on the hypersurfaces.

four spacecraft moving with one velocity, as suggested by previous studies (De Keyser et al., 2007; De Keyser, 2008). If there are only three spacecraft available with identical velocities, the resulting measurement points will inevitably lie in a hyperplane in time-space. Alternatively, if the three spacecraft have at least two kinds of velocities, the measurement points can deviate from a plane and the gradients can be estimated. In the case of $d = 2$, at least fifteen points are required and they should also not belong to a quadratic hypersurface. In this case, ten spacecraft flying in formation suffice. If there are only nine spacecraft,

then at least two velocities are needed.

     To prove these necessitates a heavy symbolic computation best suited for a computer to handle effectively. Here we demonstrate the effect brought about by different velocities of spacecraft by considering the $y$-axis in Fig. 2 (b) as a temporal axis. If there is only one spacecraft flying with a constant velocity, its trajectory in time-space forms a straight line, violating the requirement that measurement points should not lie on a linear path. To render the three non-linearly distributed measurement

points, in addition to by two spacecraft with the same velocity that results in parallel trajectories, it is also possible by a single spacecraft with changing velocity that leads to a non-linear trajectory. Similarly, treating the $z$-axis in Fig. 2 (c) as a temporal

one, the four measurement points can result from three spacecraft with the same velocity, viz. passed by three parallel lines. They can also be generated by two spacecraft with different velocities, each following a trajectory connecting a pair of points.

## 4.2 When the Requirement is Not Met

In practice, there are situations where the requirement is not met. For $d = 1$, this can occur due to instrument failures in a four-spacecraft mission or a lack of spacecraft to form a tetrahedron, resulting in only three spacecraft providing data that lie on a plane. Even in well-functioning four-spacecraft missions, orbital constraints can cause the spacecraft to be nearly coplanar at times. For $d = 2$, many current probes are distributed spherically, such as geomagnetic stations on the solid Earth or the Iridium satellite constellation in the ionosphere. The upcoming HelioSwarm mission will consist of only nine spacecraft. In future missions involving ten or more spacecraft, the same challenges faced by four-spacecraft missions can also arise. Hence, it is crucial to explore whether there exists a method to effectively leverage the available data in such cases.

The direct problem is that Eq. (14) has infinite number of solutions as the determinant of $\mathbf{R}$ becomes zero. One potential approach is to address this problem by excluding certain gradient components from the approximating polynomial (Eq. (4)) and relocating them to the truncated one (Eq. (5)). By doing so, the degrees of freedom in the problem can be adjusted to match the available measured data. However, it is important to note that randomly dropping components is not suitable, as Section 5 will demonstrate that this can lead to overwhelming errors and deterioration of all estimated $g_\alpha$ ($|\alpha| = d$). To effectively reduce the degrees of freedom in the approximation, it is necessary to consider the degrees of freedom in the distribution of measurement points, specifically the rank of $\mathbf{X}$, and the surfaces that contain the measurement points.

Suppose that there exist $N$ sets of coefficients $a_\alpha$ such that Eq. (19) is satisfied, which indicate that the measurement points lie on the intersection of $N$ distinct surfaces of degrees at most $d$, and that the rank of $X$ is $\binom{d+r}{r} - N$. Take $d = 1, r = 3$ for example. If $N = 1$ ($N = 2$), then all points lie on a surface (line) and the rank of $\mathbf{X}$ is 3 (2). By right multiplying $\mathbf{X}$ with a full rank square matrix $\mathbf{G}$, it is possible to obtain a matrix $\mathbf{X}'$ whose last $N$ columns are zeros. To put it more visually, it is possible to have

$$\mathbf{X}'_{nh} = \sum_l^{\binom{d+r}{r}} X_{nl} G_{lh} \tag{23}$$

such that

$$X' = \begin{bmatrix} X'_{11} & \cdots & \overbrace{0}^{N} & \cdots & 0 \\ X'_{21} & \cdots & 0 & \cdots & 0 \\ \vdots & \ddots & \vdots & \ddots & \vdots \\ X'_{M1} & \cdots & 0 & \cdots & 0 \end{bmatrix}. \tag{24}$$

Each of the last $N$ columns of $\mathbf{G}$ is a set of coefficients $a_\alpha$ that represents a surface that contain the measurement points. Since the process to obtain the $\mathbf{G}$ is quite involved and do not affect the scheme to calculate gradients, we shall defer the discussion until the scheme is fully revealed. Eq. (23) can be interpreted as a singular value decomposition in the form of $\mathbf{X} = \mathbf{ASB}$

where $\mathbf{S}$ is a $M \times \binom{d+r}{r}$ diagonal matrix and $\mathbf{A}$ and $\mathbf{B}$ are unitary matrices of order $M$ and $\binom{d+r}{r}$ respectively (Kincaid and Cheney, 2002). If the transformation matrix $\mathbf{G}$ in Eq. (23) is chosen to be unitary, we have $G = B^{-1}$ and $X' = AS$.

    By left multiplying Eq. (14) with $\mathbf{G}^{\mathrm{T}}$, making use of Eqs. (12) and (13), and considering the decomposition $\mathbf{I} = \mathbf{G}\mathbf{G}^{-1}$ where $\mathbf{I}$ is the identity matrix, we obtain

$$(\mathbf{X}')^{\mathrm{T}}\mathbf{W}\boldsymbol{j} = (\mathbf{X}')^{\mathrm{T}}\mathbf{W}\mathbf{X}'\mathbf{G}^{-1}\tilde{\boldsymbol{g}} \tag{25}$$

where $\boldsymbol{j}$ and $\tilde{\boldsymbol{g}}$ are column vectors. $\mathbf{G}^{-1}\tilde{\boldsymbol{g}}$ represents a recombination of the gradient components according to the distribution of measurement points. In matrix form, this equation writes:

$$N\left\{\begin{bmatrix} \boldsymbol{J}' \\ \hline 0 \\ \vdots \\ 0 \end{bmatrix} = \begin{bmatrix} \mathbf{R}' & \overbrace{\begin{matrix} 0 & \cdots & 0 \\ \vdots & \ddots & \vdots \end{matrix}}^{N} \\ 0 & \cdots & 0 & \cdots & 0 \\ \vdots & \ddots & \vdots & \ddots & \vdots \\ 0 & \cdots & 0 & \cdots & 0 \end{bmatrix} \begin{bmatrix} \boldsymbol{g}' \\ \hline (\mathbf{G}^{-1}\tilde{\boldsymbol{g}})_{\binom{d+r}{r}-N+1} \\ \vdots \\ (\mathbf{G}^{-1}\tilde{\boldsymbol{g}})_{\binom{d+r}{r}} \end{bmatrix}, \tag{26}$$

where $\boldsymbol{J}'$ and $\mathbf{R}'$ contain non-vanishing components and $\tilde{\boldsymbol{g}}'$ includes the first $\binom{d+r}{r} - N$ components of $\mathbf{G}^{-1}\tilde{\boldsymbol{g}}$. Since Eq. (23) and (26) can also be obtained by singular value decomposition, the geometrical interpretation of the existence condition of a

235 unique solution given in this study is also equivalent to the one in terms of singular values (De Keyser et al., 2007). By Eq. (26) we have separated from the last $N$ insoluble components of $\mathbf{G}^{-1}\tilde{\boldsymbol{g}}$ the soluble $\tilde{\boldsymbol{g}}'$. By solving for them from

$$\boldsymbol{J}' = \mathbf{R}'\tilde{\boldsymbol{g}}', \tag{27}$$

we can extract the maximum amount of information about gradients from the measurement points of a given distribution.

    Let us illustrate this method with a simple example when $d = 1$, $r = 3$, and all points satisfy $z = 0$. In this case, $N = 1$
and $\mathbf{X}$ itself is in the form of $\mathbf{X}'$. Thus, identity matrix can be used in place of $\mathbf{G}$ to give a set of unknowns, $\mathbf{G}^{-1}\tilde{\boldsymbol{g}} = [f(\bar{0}), \partial_x f(\bar{0}), \partial_y f(\bar{0}), \partial_z f(\bar{0})]$, which suggests that the gradient along the $z$-direction cannot be estimated while the rest can still be obtained. This is intuitive in the case of estimating linear gradients. And the problem has been addressed previously by the use of reciprocal vectors (Vogt et al., 2009). The benefit of the method here, however, comes from its general applicability in problems of all orders and for future missions that consist of more spacecraft.

At last we discuss how to obtain $\mathbf{G}$. The possible choices of $\mathbf{G}$ are infinite, since the form of $\mathbf{X}'$ is invariant upon the linear recombination of the last $N$ columns of $\mathbf{G}$ and the random replacement of the first $\binom{d+r}{r} - N$ columns as long as the resultant $\mathbf{G}$ has full rank. Among all possible $\mathbf{G}$, the most readily available one is the matrix of Gauss elimination, which we denote by $\mathbf{G}^*$. To obtain this matrix, we perform Gauss column elimination on $\mathbf{X}$ so that the resulted $\mathbf{X}'$ is triangular in its upper left. Each elementary column operation of the elimination is equivalent to the right multiplication of an elementary matrix. The
product of these elementary matrices is $\mathbf{G}^*$.

    To facilitate the error analysis in Section 5, we can also construct from the matrix of Gauss elimination a set of special $\mathbf{G}$, which we denote by $\mathbf{G}'$. The last $N$ columns of $\mathbf{G}'$ are those of the $\mathbf{G}^*$, $G'_{lh} = G^*_{lh}$ for $1 \leq l \leq \binom{d+r}{r}$ and $\binom{d+r}{r} - N < h \leq$

$\binom{d+r}{r}$. In the first $\binom{d+r}{r} - N$ columns, in addition to the rest being zeros, $\binom{d+r}{r} - N$ unities are so placed that the following two conditions are met:

1. $\mathbf{G}'$ has full rank.

2. Let the row (column) index of a unity be $i$ ($j$). If $\binom{u-1+r}{r} < i \le \binom{u+r}{r}$ for some $u$ and $\binom{v-1+r}{r} < j \le \binom{v+r}{r}$ for some $v$, then we should have $u = v$.

## 5  Analytical Error Analysis

While a unique solution can be obtained for estimating $\tilde{g}_\alpha$ and $\tilde{p}_d(\boldsymbol{x})$, the accuracy may vary significantly due to various factors. One factor that influences the accuracy is the choice of the weight matrix $W_{mn}$. If prior information about the background field $f$ and the wave field $w$ is available, it is possible to adapt the weight matrix appropriately to improve the accuracy, as suggested by (De Keyser, 2008). For general purposes, the plain form of Eq. (10) is sufficient. This form provides a reasonable balance between simplicity and effectiveness in capturing the underlying field characteristics.

Let $\mathbf{R}^{-1}$ be the inverse of $\mathbf{R}$. We multiply Eq. (14) with $\left(R^{-1}\right)_{\gamma\beta}$, sum over $\beta$, and obtain

$$\sum_{|\beta| \le d} \left(R^{-1}\right)_{\gamma\beta} \sum_{m,n} \boldsymbol{x}_m^\beta W_{mn} [p_d(\boldsymbol{x}_n + \delta\boldsymbol{x}) + p_d^+(\boldsymbol{x}_n + \delta\boldsymbol{x}) + w(\boldsymbol{x}_n + \delta\boldsymbol{x}) + \delta j] = \tilde{g}_\gamma, \tag{28}$$

where use was made of Eq. (13) and (7). According to the binomial theorem for multivariate polynomials (see Eq. (A1)), we have the decomposition

$$p_d(\boldsymbol{x} + \delta\boldsymbol{x}) = \sum_{|\alpha| \le d} g_\alpha \sum_{\bar{0} \le \lambda < \alpha} \binom{\alpha}{\lambda} \boldsymbol{x}^\lambda \delta\boldsymbol{x}^{\alpha-\lambda} + \sum_{|\alpha| \le d} g_\alpha \boldsymbol{x}^\alpha \tag{29}$$

Substituting this into Eq. (28), subtracting $g_\gamma$ from both sides, and defining the error in estimating $g_\gamma$ as

$$\delta g_\gamma \equiv \tilde{g}_\gamma - g_\gamma, \tag{30}$$

we obtain the complete expression for the error

$$\sum_{|\beta| \le d} \left(R^{-1}\right)_{\gamma\beta} \sum_{m,n}^{M} \boldsymbol{x}_m^\beta W_{mn} \left[ \sum_{|\alpha| \le d} g_\alpha \sum_{\bar{0} \le \lambda < \alpha} \binom{\alpha}{\lambda} \boldsymbol{x}_n^\lambda \delta\boldsymbol{x}^{\alpha-\lambda} \right.$$
$$\left. + p_d^+(\boldsymbol{x}_n + \delta\boldsymbol{x}) + w(\boldsymbol{x}_n + \delta\boldsymbol{x}) + \delta j \right] = \delta g_\gamma. \tag{31}$$

The terms in the brackets on the left represent errors of various origins.

Here we consider the error cased by the truncation of Taylor series, i.e. the term containing $p_d^+(\boldsymbol{x}_n + \delta\boldsymbol{x})$. Making use of Eq. (5), we express the relative truncation error in $g_\gamma$ as

$$\frac{(\delta g_\gamma)_{\mathrm{t}}}{g_\gamma} = \sum_{|\alpha| > d} \frac{g_\alpha}{g_\gamma} \sum_{|\beta| \le d} \left(R^{-1}\right)_{\gamma\beta} \sum_{m,n} \boldsymbol{x}_m^\beta W_{mn} \left(\boldsymbol{x}_n^\alpha + \delta\boldsymbol{x}\right), \tag{32}$$

It is obvious that three factors combine to make this error. The first is the ratio of higher-order coefficients $g_\alpha$ to $g_\gamma$, which is inherent to the nature of the field being estimated. This ratio can be modeled by $D^{|\gamma|-|\alpha|}$ where $D$ is the scale of the field. The second is the values of the measurement points $\boldsymbol{x}_m$ which appear in both the inverse of $R$ and the terms after the last summation sign. These values are determined by the choice of the origin and the size and configuration of measurement points. The third is the positioning error in time-space. The MMS mission consists of four spacecraft that fly in close formation. The separation among them can reach 10km at times with a relative position error of less than 100m, i.e. 1% of the separation. Since as compared to the differences in measurement points $\boldsymbol{x}_m$, $\delta\boldsymbol{x}$ is usually small, we could ignore it here for the purpose of estimating truncation errors. Then we have the error as a sum of terms at various orders

$$\frac{(\delta g_\gamma)_{\rm t}}{g_\gamma} = \sum_{|\alpha|>d} \frac{g_\alpha}{g_\gamma} q^{\#}_{\alpha\gamma} \max_m |\boldsymbol{x}_m|^{|\alpha|-|\gamma|} \tag{33}$$

where $q^{\#}_{\alpha\gamma}$ are dimensionless figures that can be calculated by comparing Eq. (33) with Eq. (32). $\#$ is used to indicate that $q^{\#}_{\alpha\gamma}$ has little physical meaning and will be replaced later.

It then is obvious that to reduce the error it is pertinent to choose the center of measurement points as the origin and so we have

$$\sum_m \boldsymbol{x}_m = \bar{0}. \tag{34}$$

Thus, Eq. (33) can be re-expressed as

$$\frac{(\delta g_\gamma)_{\rm t}}{g_\gamma} = \sum_{|\alpha|>d} \frac{g_\alpha}{g_\gamma} \frac{1}{q_{\alpha\gamma}} L^{|\alpha|-|\gamma|} \tag{35}$$

where $L$ is the characteristic dimension of the distribution of measurement points. $L$ can be modeled by the square roots of the eigenvalues of the volumetric tensor (Harvey, 1998). The volumetric tensor $\mathbf{R}$ is defined by Eq. (12) when $W_{mn} = \delta_{mn}/M$ and $|\alpha| = |\beta| = 1$. $q_{\alpha\gamma}$ are parameters to be calculated by comparing Eq. (35) with Eq. (32):

$$q_{\alpha\gamma} = \frac{L^{|\alpha|-|\gamma|}}{\sum_{|\beta|\leq d} (R^{-1})_{\gamma\beta} \sum_{m,n} \boldsymbol{x}_m^\beta W_{mn} \boldsymbol{x}_n^\alpha}. \tag{36}$$

They are determined by the distribution of measurement points. For a given characteristic scale $L$ of the points, through $q_{\alpha\gamma}$ the error of estimation can be affected by the distribution of points. Therefore, they can be termed as quality factors that indicate whether or not the distribution is sound for the estimation. The absolute value of these quality factors vary from zero to infinity, with larger value representing better quality. In particular, the quality factors of $|\alpha| = d+1$ are the most important since other quality factors correspond to higher orders of $L$.

In common cases, the accuracy of $\tilde{g}_\gamma$ is at the order of $d+1-|\gamma|$. For example, in estimating the gradients up to second order ($d = 2$), the accuracy of the linear gradients is of second order and that of quadratic gradients is of first order. This conclusion was also suggested by previous tests on synthetic data (Shen et al., 2021c). If the estimation is made up to third order ($d = 3$), the accuracy of the linear gradient could reach third order and that of the quadratic gradient becomes of second order.

The total error in Eq. (31) diminishes as the separation between measurement points is reduced, until the effects of errors in position and measurement and the effect of wave field take hold, though the latter can be mitigated by low-pass filtering. For a specific field, the errors in position and measurement and the wave field collectively set an upper limit on the accuracy achievable through the manipulation of measurement point configurations; Conversely, for a fixed set of measurement points, the accuracy depends on the comparative magnitudes of background variations versus those of the wave field and measurement errors. In practice, the total relative error can be computed from Eqs. (31) and (32) by modeling the wave field, position error, and measurement error as random variables, and by using $\tilde{g}_\alpha$ in place of $g_\alpha$ for $|\alpha| \leq d$ and $D^{|\gamma|-|\alpha|}$ in place of $g_\alpha/g_\gamma$ for $|\alpha| > d$.

We now consider the error involved in the reconstruction of field, i.e. by Eq. (15). The error is given by

$$f(\boldsymbol{x}) - \tilde{p}_d(\boldsymbol{x}) = p_d^+(\boldsymbol{x}) - \sum_{|\gamma| \leq d} \delta g_\gamma \boldsymbol{x}^\gamma. \tag{37}$$

Using Eqs. (5) and (33) we arrive at a trivial conclusion that the degree of the error is at least $d + 1$.

When the degrees of freedom in the measured data are less than the required $\binom{d+r}{r}$, the method described in Section 4.2 can be utilized. To analyze the involved error, we apply the foregoing procedures once more. we left multiply Eq. (27) with $\mathbf{R}'^{-1}$ to obtain:

$$\sum_h^{\binom{d+r}{r}-N} (R'^{-1})_{lh} \sum_{m,n}^M (X'^{\mathrm{T}})_{hm} W_{mn} [p_d(\boldsymbol{x}_n + \delta\boldsymbol{x}) + p_d^+(\boldsymbol{x}_n + \delta\boldsymbol{x}) + w(\boldsymbol{x}_n + \delta\boldsymbol{x}) + \delta j] = \tilde{g}_l', \tag{38}$$

By defining

$$\delta g_l' \equiv \tilde{g}_l' - g_l', \tag{39}$$

we have the following expression for it:

$$\sum_h^{\binom{d+r}{r}-N} (R'^{-1})_{lh} \sum_{m,n}^M (X'^{\mathrm{T}})_{hm} W_{mn} \left[ \sum_{|\alpha| \leq d} g_\alpha \sum_{\bar{0} \leq \lambda < \alpha} \binom{\alpha}{\lambda} \boldsymbol{x}_n^\lambda \delta\boldsymbol{x}^{\alpha-\lambda} + p_d^+(\boldsymbol{x}_n + \delta\boldsymbol{x}) + w(\boldsymbol{x}_n + \delta\boldsymbol{x}) + \delta j \right] = \delta g_l'. \tag{40}$$

The relative error caused by truncation is given by

$$\frac{(\delta g_l')_\mathrm{t}}{g_l'} = \sum_{|\alpha| > d} \frac{g_\alpha}{g_l} \sum_h^{\binom{d+r}{r}-N} (R'^{-1})_{lh} \sum_{m,n}^M (X'^{\mathrm{T}})_{hm} W_{mn} (\boldsymbol{x}_n^\alpha + \delta\boldsymbol{x}), \tag{41}$$

When the $\mathbf{G}'$ presented in Section 4.2 are used for $\mathbf{G}$, the elements in $\mathbf{R}'$ and $\mathbf{X}'$ are at the same order of $L$ as are the elements of $\mathbf{R}$ and $\mathbf{X}$. Thus, the error can be expressed as

$$\frac{(\delta g_l')_\mathrm{t}}{g_l'} = \sum_{|\alpha| > d} \frac{g_\alpha}{g_l} \frac{1}{q_{\alpha l}'} L^{|\alpha|-u}, \quad \text{if } \binom{u-1+r}{r} < l \leq \binom{u+r}{r}, \tag{42}$$

where the quality factor is given by

$$q_{\alpha l}' = \frac{L^{|\alpha|-u}}{\sum_h^{\binom{d+r}{r}-N} (R^{-1})_{lh} \sum_{m,n}^M (X'^{\mathrm{T}})_{hm} W_{mn} \boldsymbol{x}_n^\alpha}, \quad \text{if } \binom{u-1+r}{r} < l \leq \binom{u+r}{r}. \tag{43}$$

Therefore, this method designed for cases when measurement points are not well distributed has good accuracy.

## 6   Summary and Discussion

The techniques for calculating linear gradients of general physical fields and quadratic gradients of magnetic fields using four-point measurements have been widely applied in the context of multispacecraft missions to advance our understanding of space plasma. However, there are also important quantities and processes associated with the quadratic gradients of other fields that warrant further exploration. For instance, the gradients of velocity play a crucial role in determining fundamental quantities such as viscosity and energy dissipation rate. Overall, the statics and dynamics of physical fields in space are interrelated through their gradients. As the number of spacecraft in a constellation continues to increase, it is helpful to explore and prepare for future missions multipoint techniques that rely on more points to estimate quadratic and higher-order gradients.

In summary, we have analytically explored the general method to estimate gradients of fields in space based on multipoint measurement. Regarding the feasibility of estimation, a general conclusion is that to estimate the complete gradients up to $d$th degree using simultaneous measurement, $\binom{d+3}{3}$ spacecraft are needed and these spacecraft should not lie on a $d$th-order surface in space. In particular, at least ten points that are not on a second-order surface are needed to estimate both linear and quadratic gradients. To address the negative effects caused by poor synchronization among spacecraft in a large constellation and to estimate the additional temporal gradients of a field, time series needs to be taken into account and it is necessary to have at least $\binom{d+4}{4}$ measurement points that do not lie on a $d$th-order hypersurface in time-space. For linear gradients, these measurement points can be provided by a constellation of four spacecraft having the same velocity or of three spacecraft whose velocities have at least two kinds. For quadratic gradients, ten co-moving spacecraft are sufficient. It is also possible to reduce one spacecraft by adding one more velocity. In situations where the measured data lacks degrees of freedom due to an ill configuration of spacecraft, which may include a shortage of spacecraft, it becomes necessary to invoke a transformation in order to estimate the gradient components to the best extent possible.

Regarding the accuracy, we have analytically proven that in an estimation of gradients up to $d$th order, the order of accuracy of the $a$th-order gradients is at least $d+1-a$. We have also provided quality factors $q_\alpha$ to judge the distribution of measurement points and the spacecraft configuration in a constellation. In addition, a method for estimating errors in real time has also been presented.

The results obtained offer valuable insights for the development of multipoint techniques that rely on gradients of physical fields. Additionally, they hold significance for the future design of multispacecraft missions aimed at studying physics associated with quadratic or higher-order gradients.

The current study primarily focuses on approximating a single scalar physical field in space. It treats vector and tensor as aggregations of multiple independent scalar fields. In practice, the constituents of a vector field can be interrelated, as exemplified by the divergence-free condition $\nabla \cdot \boldsymbol{B} = 0$ for magnetic field. The gradients of different fields can also be subject to various physical formula. For instance, the zeroth-order gradients of electric current and first-order gradients of magnetic field are correlated by $\nabla \times \boldsymbol{B} = \mu_0 \boldsymbol{j}$. Beyond these linear constraints, non-linear constraints exist as well. The gradients of entropy $s$ and velocity $\boldsymbol{v}$ of an isentropic flow satisfy $\partial_t s + \boldsymbol{v} \cdot \nabla s = 0$. To incorporate all conceivable constraints into the current framework is challenging and will be explored in the future. However, when only a limited number of constraints, such

as the sole divergence-free condition for magnetic field, are taken into account, the existence condition remains unchanged for a complete solution concerning the configuration of measurement points in time-space.

From the numerical point of view, the matrix $\mathbf{R}$ (Eq. (12)) is likely to be ill-conditioned, for it is the weighted product of two
Vandermonde matrices. This together with the limited resolution of measurement puts a limitation on the practicability of the technique on approximating to higher orders by solving Eq. (14), though the framework is in principle applicable to all orders. Thus for higher order approximations, it is necessary to verify in Eq. (31) that the error resulting from the multiplication of $\delta j$ with the terms outside the square brackets is not substantial. As for quadratic gradients, previous simulations have verified the feasibility, reliability, and accuracy of the technique (Shen et al., 2021c).

## 375   **Appendix A: Multi-index Notation**

Here we list the properties of multi-index notation tailored for multivariate functions (Riachy et al., 2011).

Let $\alpha = (\alpha_1, \ldots, \alpha_r)$ be an $r$-tuple of non-negative integers $\alpha_i, i = 1, \ldots, r; i, r \in \mathbb{N}$. $\alpha$ is called a multi-index. The symbol in bold $\boldsymbol{x}$ denotes a vector in $\mathbb{R}^r$. As for a time-space, $r = 4$.

For multi-indices $\alpha, \beta \in \mathbb{N}^r$ the following properties are either defined or deduced.

1. Componentwise sum and difference: $\alpha \pm \beta = (\alpha_1 \pm \beta_1, \ldots, \alpha_r \pm \beta_r)$.

2. Partial order $\alpha \leq \beta \Leftrightarrow \alpha_i \leq \beta_i, \forall i \in \{1, \ldots, r\}$. $\alpha = \beta \Leftrightarrow \alpha_i = \beta_i, \forall i \in \{1, \ldots, r\}$.

3. Given $\boldsymbol{x} = (x_1, \ldots, x_r) \in \mathbb{R}^r$, we have that $\boldsymbol{x}^\alpha = x_1^{\alpha_1} \cdots x_r^{\alpha_r}$.

4. The total degree of $\boldsymbol{x}^\alpha$ is given by $|\alpha| = \alpha_1 + \cdots + \alpha_r$.

5. Factorial: $\alpha! = \alpha_1! \cdots \alpha_r!$.

6. Binomial coefficient: $\binom{\alpha}{\beta} = \binom{\alpha_1}{\beta_1} \cdots \binom{\alpha_r}{\beta_r}$

7. $\bar{b} = (b, \ldots, b), b \in \mathbb{N}, \bar{b} \in \mathbb{N}^r$

8. Higher-order partial derivative $\partial^\alpha \equiv \partial_1^{\alpha_1} \cdots \partial_r^{\alpha_r}$ where $\partial_i^{\alpha_i} \equiv \frac{\partial^{\alpha_i}}{\partial x_i^{\alpha_i}}$. $\partial^\alpha f \equiv f_{,\alpha}$.

9. Denote by $1_i \in \mathbb{N}^r$ the multi-index with zeros for all elements except the $i$th one i.e. $1_i = (0, \ldots, 0, 1, 0, \ldots, 0)$.

10. The tensor product of 2 vectors $\boldsymbol{u}, \boldsymbol{v} \in \mathbb{R}^r$ is defined by $\boldsymbol{u} \otimes \boldsymbol{v} = (u_1 \boldsymbol{v}, \ldots, u_r \boldsymbol{v}) \in \mathbb{R}^{r^2}$.

11. Binomial theorem:

$$(\boldsymbol{x} + \boldsymbol{y})^\alpha = \sum_{\bar{0} \leq \beta \leq \alpha} \binom{\alpha}{\beta} \boldsymbol{x}^\beta \boldsymbol{y}^{\alpha - \beta} \tag{A1}$$

## Appendix B: Scaling Coordinates

In addition to the scaling of temporal coordinate by a characteristic speed to obtain $x_0 = vt$, it has been suggested that scaling on the three spatial coordinates can be invoked to further improve accuracy when the spatial variations of the physical fields are highly anisotropic (De Keyser et al., 2007). A recent observational study (Liu et al., 2022) showed that the ratio of the characteristic scale parallel to magnetic fields over perpendicular scales are roughly 2:1 for solar wind and magnetosheath plasmas. A corresponding scaling as such can be applied to the spatial coordinates of measurement points before calculating the matrix $\mathbf{R}$ and the vector $\boldsymbol{J}$ and solving for the gradients from Eq. (14).

*Author contributions.* YZ conceived of the study and performed the investigation, with contributions from CS. YZ wrote the paper. CS reviewed and edited the paper.

*Competing interests.* No competing interests.

*Acknowledgements.* This work was supported by the National Natural Science Foundation of China (Grants No. 42130202) and the National Key Research and Development Program of China (Grant No. 2022YFA1604600).

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
