# Peer review of "Estimating Gradients of Physical Fields in Space"

_Annales Geophysicae, 2023_

## Author Comment (AC2)

**I. RESPONSE TO REVIEWER 2**

RC2: 'Comment on angeo-2023-30', Anonymous Referee #2, 16 Nov 2023 The authors explore future multipoint techniques for constellation missions to estimate gradients of physical quantities. The analytical theory is well developed and comprehensive. I only have minor comments on the text as presented. The manuscript, however, contains no figures which this reviewer feels would greatly aid interpretation by readers who are less mathematical in their thinking and more visual. A few simple diagrams demonstrating the concepts and findings would greatly complement the existing text. If some figures are added and the minor points below are addressed, I would recommend publication.

Reply: We thank the reviewer for this suggestion. Diagrams will be provided to aid the reading by a wide audience.

Line 30: Change "of reconnections" to "in reconnection"

Reply: to be corrected

Lines 31-32: This states the reconstruction avoids assumptions, however, the underlying assumptions about the forms of gradients are omitted, e.g. that they are relatively consistent over the scales of the spacecraft separation.

Reply: Yes. This can be viewed as an assumption, as suggested the reviewer, for the method to successfully estimate the presupposed gradients of some physical field. It may also be viewed as a result of the method, for the method always estimates gradients over the spacecraft separation.

Lines 56-59: It would be good to explicitly mention that in practical applications measurements include noise which may then affect estimates of gradients.

Reply: Thanks. Errors of various origins will be mentioned explicitly here.

Line 70: change to "dipole (and higher-order moments)"

Reply: to be corrected

Lines 71-72: The magnetosheath is highly non-uniform over the scale of its thickness, so please be specific over what sorts of distances you are referring to.

Reply: The distinction and hence categorization between uniformity and variation are in principle artificial. In the text we were referring to the 100-200 MK background in the subsolar magnetosheath (See, e.g. Figs 2-6 of Dimmock et al. 2015 doi.org/10.1002/2014JA020734). Admittedly, the magnetosheath is highly turbulent, especially downstream of quasi-parallel shocks. Since the example of magnetosheath is not crucial for the present study, to make things simple we will remove this part.

Line 74: Are the wave fields really waves or just residuals? You mention they must have smaller scales, referring to their physical size, but do they not also need to have smaller amplitude fluctuations?

Reply: They are real waves whose amplitude can be large as compared with the variation of background field at the scale of spacecraft separation. If large-amplitude waves are retained during the estimation of gradients, the error caused by them could overwhelm the result. The estimation of the associated error is contained in Eq. 31. More discussion about waves will be added. One possible approach is to filter them before the estimation of gradients.

Line 77: It would be good to mention if the speed v needs to be chosen to be the same for all measurement points or if it can be allowed to vary.

Reply: Thanks. It will be mentioned that v is a constant to make figures in the matrix R (Eq. 12) of similar magnitude.

Line 80 (and throughout): "A" needs to change to "Appendix A"

Reply: to be corrected

Line 147: "a algebraic" change to "an algebraic"

Reply: will be corrected

Lines 197-203: This is almost identical to the previous paragraph, remove.

Reply: to be corrected

---

## Author Response (AR1)

Thank you again for your suggestions and comments, which we find very helpful for improving the manuscript. The lining number mentioned in this Response To Reviewers refers to the updated manuscript with tracking.

**I.  RESPONSE TO REVIEWER 1**

RC1: 'Comment on angeo-2023-30', Anonymous Referee #1, 10 Nov 2023

**General comments**

This paper discusses the solution of the least-squares system that stems from a multi-dimensional Taylor series approximation of a scalar field. The authors foresee the inclusion of the time dimension in their analysis. The paper does a nice job of pointing out the opportunities and some of the difficulties of applying such least-squares gradient computation techniques to situations where a larger number of spacecraft is available and/or where one attempts to assess the higher-than-linear gradients. The existence conditions are interpreted in a geometric way, which is helpful. The paper contains new ideas and is at the forefront of research.

The paper is well-structured. There is a good overview of the relevant literature. The paper also puts the work properly into context. The conclusions are clear. The paper length is appropriate.

I have a few suggestions for improving the presentation of the material and I also have a few questions, see specific comments below.

There are several language and typographical issues in the manuscript; see technical corrections listed below. The paper would benefit from being thoroughly reread once more.

The paper will likely be suitable for publication after minor revision.

Reply: We thank the reviewer for these constructive comments. We will carefully review the paper and address any technical issues to the best of our ability.

**Specific comments**

The authors discuss the weighted least-squares problem that arises when trying to fit a polynomial approximation to the observations of a scalar field at multiple points. This is, in general, an overdetermined problem. The solution of such a problem is well-known standard numerical mathematics: There is the theory of the "generalized inverse" of an overdetermined system and the use of the singular value decomposition to solve the system. None of that math, however, is referred to explicitly in the paper, while the paper reflects that math in the specific context of gradient computation. For instance, matrix (26) shows the singular value decomposition of the original matrix, which in the case discussed here is rank-deficient, reflecting the existence conditions for a solution in terms of the singular values as formulated by De Keyser et al. 2007. It would be extremely useful to highlight throughout the entire manuscript how the findings reflect this standard mathematical approach which many readers have as background knowledge.

Reply: We thank the reviewer for this suggestion. Indeed, Eq. 25 and 26 can also be obtained from singular value decomposition (SVD), and Eq. 23 can be viewed as a partial SVD. A fully fledged SVD of the matrix $X$ defined by Eq. 18 will be $X = ASB$ where $S$ is a $M \times \binom{d+r}{r}$ diagonal matrix and $A$ and $B$ are unitary matrices of order $M$ and $\binom{d+r}{r}$ respectively. If the transformation matrix $G$ in Eq. 23 is chosen to be unitary, we have, in terms of SVD, $G = B^{-1}$ and $X' = AS$. In the text, we previously omitted this interpretation to avoid the additional mathematical concept to the ease of a wide audience who might not be familiar with it. However, mentioning this concept can indeed facilitate technical readers and also help build connection between the existence condition given by De Keyser et al 2007 and the geometrical interpretation here. We have added citations to the textbook Numerical Analysis Mathematics of Scientific Computing by David Kincaid and Ward Cheney to reflect this concept and citations to previous studies about the existence conditions in the revised manuscript. Please see lines 232–234 and 241–244.

The authors focus on the basic ALQG system. They argue that the approach utilizing gradients avoids having to make specific physical assumptions. However, in many practical cases there are physical constraints such as the divergence-free nature of the magnetic field. This is briefly touched upon at the beginning of section 2. As the authors suggest one can apply the scalar field approach to each of the vector field components, but that does not incorporate the constraint yet. In section 4.2 the authors propose that one can drop some gradient components, which is a (restricted) form of geometric constraints. How can the proposed technique incorporate more generic physical and geometric constraints? How do the conclusions of the present paper generalize to the situation where such constraints can be applied?

Reply: We thank again for this comment. While the present study is in principle devoted to explore the most general geometric constraints (in time-space) resulted from the distribution of measurement points, thus contributing to future orbital design of multi-spacecraft missions, it is very useful and important to incorporate, especially in the case of rank deficiency caused by an inappropriate configuration of measurement points as discussed in Section 4.2, more physical and geometric constraints, if possible, to add constituent equations to Eq. 26 to increase the rank of $R'$ and hence to reduce the number of components in $G^{-1}\tilde{g}$ that need to be dropped. However, to give a general solution

to this problem is difficult, as on the one hand different forms of constraints have different results and on the other hand non-linearity may be introduced by some particular forms of constraints.

In the case of multiple scalar fields with $i$ denoting the $i$th field, we could extend Eqs. 8, 9, 12, 13, and 14 to

$$s_{mi} = j_{mi} - p_{di}(\mathbf{x}_m) \tag{1}$$

$$S = \sum_i^3 \sum_M^{m,n} s_{mi} W_{mn} s_{ni} \tag{2}$$

$$R_{\beta\alpha} \equiv \sum_{m,n}^M \mathbf{x}_m^\beta W_{mn} \mathbf{x}_n^\alpha, \tag{3}$$

$$J_{\beta i} \equiv \sum_{m,n}^M \mathbf{x}_m^\beta W_{mn} j_{ni}. \tag{4}$$

$$J_{\beta i} = \sum_{|\alpha| \leq d} R_{\beta\alpha} \tilde{g}_{\alpha i}, \tag{5}$$

For each $i$, Eq.(5) consists of $\binom{d+r}{r}$ equations and unknowns. We could combine these equations for all components together. First we define the order of multi-index

$$\alpha \prec \beta \text{ if } |\alpha| < |\beta| \text{ or when } |\alpha| = |\beta| \text{ there exists } n \leq r \text{ such that } \alpha_m = \beta_m; m \leq n \text{ and } \alpha_n > \beta_n \tag{6}$$

and accordingly an order function for multi-index $o_{dr} : \mathbb{N}^r \mapsto \mathbb{N}$ for given $d$ and $r$. For example,

$$o_{23}((0,0,0)) = 1, \quad o_{23}((1,0,0)) = 2, o_{23}((0,2,0)) = 8 \tag{7}$$

We then define, by using the inverse function $o^{-1}(\cdot)$ where the subscripts $dr$ are omitted for simplicity,

$$\mathcal{R}_{qp} = \begin{cases} R_{\alpha\beta}, \text{where } \alpha = o^{-1}(q \bmod \binom{d+r}{r}), \beta = o^{-1}(p \bmod \binom{d+r}{r}) & \text{if } q - q \bmod \binom{d+r}{r} = p - p \bmod \binom{d+r}{r} \\ 0, & \text{otherwise} \end{cases} \tag{8}$$

$$\mathcal{J}_q = J_{\alpha i}, \text{where } \alpha = o^{-1}(q \bmod \binom{d+r}{r}), \ i = \left(q - q \bmod \binom{d+r}{r}\right) / \binom{d+r}{r} \tag{9}$$

$$\tilde{\mathcal{G}}_q = \tilde{g}_{\alpha i}, \text{where } \alpha = o^{-1}(q \bmod \binom{d+r}{r}), \ i = \left(q - q \bmod \binom{d+r}{r}\right) / \binom{d+r}{r} \tag{10}$$

Visually, the matrix $\mathcal{R}$ is given by

$$\mathcal{R} = \begin{bmatrix} \mathbf{R} & & & \\ & \mathbf{R} & & \\ & & \mathbf{R} & \\ & & & \ddots \end{bmatrix} \tag{11}$$

Thus Eq.(5) for all $i$ becomes

$$\mathcal{J}_q = \sum_q R_{qp} \tilde{\mathcal{G}}_p, \tag{12}$$

Let us first consider linear constraints for $l$ scalar fields in the general form

$$\sum_q^{l\binom{d+r}{r}} a_q \mathcal{G}_q = 0 \tag{13}$$

where $a_q$ are constants. Ordering magnetic field components $B_i$ following $i = 1, 2, 3$ ($l = 3$) and taking $d = 1, r = 3$, the condition $\nabla \cdot \mathbf{B} = 0$ gives non-vanishing coefficients $a_2 = a_7 = a_{12} = 1$. For simplicity, let us consider the case of $l = 2, d = 1, r = 2, a_2 = 1, a_5 = -1$. Eliminating $\mathcal{G}_2$ by means of Eq.(13) in the total error and minimizing the error

with respect to $\mathcal{G}_q$ for $1 \leq q \leq 6$, the matrix $\mathcal{R}$ is given by

$$\mathcal{R} = \begin{bmatrix} \mathcal{R}_{11} & 0 & \mathcal{R}_{13} & 0 & \mathcal{R}_{15} & 0 \\ 0 & 0 & 0 & 0 & 0 & 0 \\ \mathcal{R}_{31} & 0 & \mathcal{R}_{33} & 0 & \mathcal{R}_{35} & 0 \\ 0 & 0 & 0 & \mathcal{R}_{44} & \mathcal{R}_{45} & \mathcal{R}_{46} \\ \mathcal{R}_{51} & 0 & \mathcal{R}_{53} & \mathcal{R}_{54} & \mathcal{R}_{55} & \mathcal{R}_{56} \\ 0 & 0 & 0 & \mathcal{R}_{64} & \mathcal{R}_{65} & \mathcal{R}_{66} \end{bmatrix} \tag{14}$$

This matrix can be decomposed according to $\mathcal{R} = \mathcal{X}^{\mathrm{T}} \mathcal{W} \mathcal{X}$ to give

$$\mathcal{X} = \begin{bmatrix} 1 & 0 & x_{12} & 0 & x_{11} & 0 \\ 1 & 0 & x_{22} & 0 & x_{21} & 0 \\ 1 & 0 & x_{32} & 0 & x_{31} & 0 \\ \vdots & \vdots & \vdots & \vdots & \vdots & \vdots \\ 0 & 0 & 0 & 1 & x_{11} & x_{12} \\ 0 & 0 & 0 & 1 & x_{21} & x_{22} \\ 0 & 0 & 0 & 1 & x_{31} & x_{32} \\ \vdots & \vdots & \vdots & \vdots & \vdots & \vdots \end{bmatrix} \tag{15}$$

where

$$\mathcal{W} = \begin{bmatrix} \mathbf{W} & \\ & \mathbf{W} \end{bmatrix} \tag{16}$$

If there exists a line

$$b_{(0,0)} + b_{(1,0)} x_1 + b_{(0,1)} x_2 = 0 \tag{17}$$

such that all points $\mathbf{x}_m$ lie on this line, we have rank $\mathcal{X} \leq 4$. To see this, we first prepare a matrix from Eq.(16) by adding the fourth column to the first and the last column to the third. Then combining the first, third, and fifth column with the coefficient given in Eq.(17) gives a column of zero. Thus, according to the procedure given in Section 4.2 in the manuscript, we have to drop at least one component of gradient. The result can be easily generalized to the most general case of Eq.(13).

Non-linear constraints are difficult to tackle. Taking the example of estimating the linear gradients of velocity and entropy of an one-dimensional isentropic flow, we order the two field as $(v, s)$. The constraint is

$$\mathrm{d}s/\mathrm{d}t = \partial_0 s + v \partial_1 s = 0 \tag{18}$$

or

$$\mathcal{G}_5 + \mathcal{G}_1 \mathcal{G}_6 = 0 \tag{19}$$

By eliminating $\mathcal{G}_5$ in the total error, and following the procedure given in the manuscript, we obtain the matrix

$$\mathcal{X} = \begin{bmatrix} 1 & x_{10} & x_{11} & 0 & 0 & 0 \\ 1 & x_{20} & x_{21} & 0 & 0 & 0 \\ 1 & x_{30} & x_{31} & 0 & 0 & 0 \\ \vdots & \vdots & \vdots & \vdots & \vdots & \vdots \\ -\mathcal{G}_{16} & 0 & 0 & 1 & 0 & x_{11} \\ -\mathcal{G}_{26} & 0 & 0 & 1 & 0 & x_{21} \\ -\mathcal{G}_{36} & 0 & 0 & 1 & 0 & x_{31} \\ \vdots & \vdots & \vdots & \vdots & \vdots & \vdots \end{bmatrix} \tag{20}$$

Since the unknown $\mathcal{G}_6$ present in the matrix $\mathcal{X}$ and $\mathcal{R}$, to solve Eq.(12) the method of iteration need to be adopted, that is, assigning a set of arbitrary values to the unknowns $\mathcal{G}_p$, calculating the matrix $\mathcal{R}$ and its inverse, multiplying $\mathcal{J}$ with $\mathcal{R}^{-1}$ to get a new set of values for $\mathcal{G}_p$, calculating $\mathcal{R}$... and so on, until the values of $\mathcal{G}_p$ converge. In the present example of one-dimensional isentropic flow, $\mathcal{G}_6$ is a constant in time-space, thus giving $\mathcal{G}_{16} = \mathcal{G}_{26} = \mathcal{G}_{36} = \ldots$.

Hence, in analogy with the previous linear example, if the measurement points lie on a line in time-space, we could deduce that the rank of Eq.(20) is less than five, such that one component need to be dropped following again the procedure given in.

From the discussion about linear and non-linear constraints, we can see that a limited number of additional physical constraints does not alter the existence condition in terms of measurement point distribution, in particular the geometrical configuration of a spacecraft constellation. Even if $\nabla \cdot \mathbf{B}$ is taken into account, in order to ensure the existence of the most complete solution for the gradients of magnetic field, the spacecraft should not lie on a second-order surface. However, as a major result of the paper by *Shen et al.* (2021), we can incorporate more physical fields, such as electric current, and the associated constraints that result from physical formula and assumptions, to allow the spacecraft to lie on second-order surfaces, but not on any first-order surface. This result is specific to magnetic field and cannot be generalized to other fields. In the manuscript we have added a brief discussion about this problem. Please see lines 368–376.Maybe in the future we could explore if there is any general framework to contain them all. We thank the reviewer again for this inspiring comment.

The numerical conditioning of the ALQG system depends on the scaling of the variables. The authors use the spatial coordinates as such, and the time coordinate multiplied by a characteristic speed. What if the spatial variations are very anisotropic, as is often the case in magnetic field dominated situations? Note that this is related to the question of geometric constraints and the "homogeneity scales" introduced by De Keyser et al. 2007.

Reply: Yes, the spatial variation of physical fields in space can be anisotropic. To tackle this, "homogeneity scales", which can be interpreted as linear scales, for different spatial dimensions can be utilized to improve accuracy in estimating linear gradients. And in estimating gradients of higher order, more scales can also be introduced similarly. De Keyser et al. 2007 used 4:1 for parallel over perpendicular scales. A recent observational study (*Liu et al.*, 2022) showed the ratio being roughly 2:1 for solar wind (Figure 2 of the paper)and magnetosheath (Figure 3 of the paper) plasmas. We have added a section in appendix to discuss this problem. Please see lines 81–83

The authors mention the problem of error estimation. How do they view/compare their approach with the one proposed by De Keyser et al. 2008, where the effects of measurement errors and approximation errors (in space and time) are combined?

Reply: The approach proposed by *De Keyser* (2008) includes the estimation of both errors to generate a weight matrix, which can be used to improve accuracy. His estimation of measurement error is straightforward. The estimation of approximation error is also meticulously and finely developed. While we believe this approach can work well for linear gradients, it is not immediately clear to us how to extend this approach to the case of higher orders. The method of estimation given in this paper is in principle a direct one, and can be used for error estimation at all orders. In addition, the effects of distribution of measurement points are taken into account not only in the estimation of approximation error (Eq. 32) but also for the measurement errors (Eq. 31, estimation error caused by measurement error can be estimated by multiplying $\delta j$ with the terms outside the square brackets.)

Matrix (20) is of a form that is known to be likely ill-conditioned (a Vandermonde matrix) – admittedly, this ill-conditioning is more pronounced as the polynomial approximation degree becomes higher; for a degree 2 the situation is not so bad yet. But perhaps this deserves a word of caution: the technique is in principle applicable to higher-degree approximations, but in practice there are clear limitations also from the numerical point of view.

Reply: We agree with the reviewer. A caveat has been provided in the revised manuscript. Please see line 377–382. While the era of degree 2 is approaching, the time of degree 3 and higher remain not clearly visible.

On line 267: For constellations with small spacecraft separation distances, the positioning error may become considerable. A word of caution would be welcome here.

Reply: Thanks. Discussion has been added to lines 289–292.

It is not clear to why the authors introduce the wave field w. No specific conclusions are drawn in the error analysis of section 5 about this field. I therefore believe that the authors could just think of the wave field as part of the scalar field that is to be modelled. The end result would simply be to remove w from the formalism and thus simplifying it. I leave this to the authors to judge, but when keeping w, then an explicit discussion of its role in the error analysis in section 5 would be welcome.

Reply: We thank the reviewer for pointing out this. Clarification has been added. Please see lines 314–319.The total measurement consists of, in addition to measurement error, physical fields at various scales, which include, from small scale to large scale, small wave field, background fields of low-order polynomials, background fields of high-order polynomials. The magnitude of the wave field is assumed to be known a priori and could be modeled in a similar way as is measurement error. The truncation error decreases with decreasing separation between measurement points. The errors caused by measurement error and wave field behave oppositely so as to set a lower limit to the maximum accuracy that is achievable by reducing $|\mathbf{x}|$.

**Technical corrections**

Throughout the text: replace "A" by "Appendix A" to refer to the appendix.

Reply: corrected.

Throughout the text: "Keyser" -> "De Keyser"; the references on line 400-404 should read as follows: "De Keyser, J.: …"

Reply: corrected.

30: "of reconnections" -> "the reconnection region"

Reply: In a steady global picture of the magnetosphere, key regions include the reconnection region for example in the magnetotail. Here we are inclined to the key regions being those important parts of a physical process such as a reconnection process. We have changed "of reconnections" to "in reconnection". Please see line 30.

35: "The algorithm" -> "An algorithm"

Reply: corrected.

39: "As if" -> "If"

Reply: corrected.

46: "point distribution" -> "the point distribution"

Reply: corrected.

48 and later: notation $f_{,\alpha}$ is strange; shouldn't this be $f'_\alpha$ ?

Reply: Perhaps line 84 and later? As referenced in line need line number and given in Appendix A, this is defined by $f_{,\alpha} \equiv \partial^\alpha \equiv$. The comma in subscript represents partial differentiation. Such notations are commonly used in tensor analysis, differential geometry/manifold, and the theory of general relativity. They are very simple in form, easy to use, and in particular, well-suited for the study of gradients of various order in the space of indefinite dimensions (3 for space and 4 for time-space).

150: "Similar result has" -> "A similar result has" or "Similar results have"

Reply: corrected.

152: "be on a surface" -> better "lie on a surface"?

Reply: corrected.

163: I do not think "great" has the connotation that you want in this sentence. Better "important"?

Reply: corrected.

163: "desgins" -> "design"

Reply: corrected.

172: "sphere" -> "the sphere" or "a sphere"

Reply: corrected.

204: "exists" -> "exist"

Reply: corrected.

235: "To the ease of" -> better "To facilitate the"?

Reply: corrected.

245: "an unique" -> "a unique"

Reply: corrected.

310: "have" -> "has"

Reply: corrected.

**II.  RESPONSE TO REVIEWER 2**

RC2: 'Comment on angeo-2023-30', Anonymous Referee #2, 16 Nov 2023 The authors explore future multipoint techniques for constellation missions to estimate gradients of physical quantities. The analytical theory is well developed and comprehensive. I only have minor comments on the text as presented. The manuscript, however, contains no figures which this reviewer feels would greatly aid interpretation by readers who are less mathematical in their thinking and more visual. A few simple diagrams demonstrating the concepts and findings would greatly complement the existing text. If some figures are added and the minor points below are addressed, I would recommend publication.

Reply: We thank the reviewer for this suggestion. Two figures, together with associated text in lines 109–110 and 192–199, have been provided to aid the reading by a wide audience.

Line 30: Change "of reconnections" to "in reconnection"

Reply: Changed.

Lines 31-32: This states the reconstruction avoids assumptions, however, the underlying assumptions about the forms of gradients are omitted, e.g. that they are relatively consistent over the scales of the spacecraft separation.

Reply: Yes. This can be viewed as an assumption for the method to successfully estimate the presupposed gradients of some physical field. It may also be viewed as a result of the method, for the method always estimates gradients over the spacecraft separation. We have changed the sentence to reflect this limitation. Please see lines 32–34.

Lines 56-59: It would be good to explicitly mention that in practical applications measurements include noise which may then affect estimates of gradients.

Reply: Thanks. We have changed the last sentence to "In practical applications, measurements include noise which may also affect estimates of gradients. It is therefore crucial to develop a reliable method for estimating and quantifying errors of various origin." Please see lines 62–63.

Line 70: change to "dipole (and higher-order moments)"

Reply: Changed.

Lines 71-72: The magnetosheath is highly non-uniform over the scale of its thickness, so please be specific over what sorts of distances you are referring to.

Reply: The distinction and hence categorization between uniformity and variation are in principle artificial. In the text we were referring to the 100-200 MK background in the subsolar magnetosheath (See, e.g., Figs 2-6 of *Dimmock et al.* (2015)). Admittedly, the magnetosheath is highly turbulent, especially downstream of quasi-parallel shocks. Since the example of magnetosheath is not crucial for the present study, to make things simple we have removed this part.

Line 74: Are the wave fields really waves or just residuals? You mention they must have smaller scales, referring to their physical size, but do they not also need to have smaller amplitude fluctuations?

Reply: They are real waves whose amplitude can be large as compared with the variation of background field at the scale of spacecraft separation. If large-amplitude waves are retained during the estimation of gradients, the error caused by them could overwhelm the result. If the method is not to estimate the gradients of these wave fields, it is preferable to apply a filter to the data. The estimation of the associated error is contained in Eq. 31. More discussion about waves has been added to lines 314–319

Line 77: It would be good to mention if the speed v needs to be chosen to be the same for all measurement points or if it can be allowed to vary.

Reply: Thanks. This is presented more explicitly in lines 81–82 and 105.

Line 80 (and throughout): "A" needs to change to "Appendix A"

Reply: Corrected.

Line 147: "a algebraic" change to "an algebraic"

Reply: Corrected

Lines 197-203: This is almost identical to the previous paragraph, remove.

Reply: Removed.
* * *
De Keyser, J., Least-squares multi-spacecraft gradient calculation with automatic error estimation, *Annales Geophysicae*, *26*(11), 3295–3316, doi:10.5194/angeo-26-3295-2008, 2008.

Dimmock, A. P., K. Nykyri, H. Karimabadi, A. Osmane, and T. I. Pulkkinen, A statistical study into the spatial distribution and dawn-dusk asymmetry of dayside magnetosheath ion temperatures as a function of upstream solar wind conditions, *Journal of Geophysical Research: Space Physics*, *120*(4), 2767–2782, doi:10.1002/2014JA020734, 2015.

Liu, Y. Y., Z. Wang, G. Chen, Y. Yu, Z. Z. Guo, and X. Xiong, Testing the linearity of vector fields in cold and dense space plasmas, *The Astrophysical Journal*, *929*(2), 155, doi:10.3847/1538-4357/ac5d4b, 2022.

Shen, C., et al., Nonlinear magnetic gradients and complete magnetic geometry from multispacecraft measurements, *J. Geophys. Res.: Space Phys.*, *126*(8), e2020JA028,846, doi:10.1029/2020JA028846, 2021.